# Modelling hCDKL5 Heterologous Expression in Bacteria

**DOI:** 10.3390/metabo11080491

**Published:** 2021-07-28

**Authors:** Marco Fondi, Stefano Gonzi, Mikolaj Dziurzynski, Paola Turano, Veronica Ghini, Marzia Calvanese, Andrea Colarusso, Concetta Lauro, Ermenegilda Parrilli, Maria Luisa Tutino

**Affiliations:** 1Department of Biology, University of Florence, Sesto F.no Florence, 50019 Florence, Italy; stefano.gozi@stud.unifi.it; 2Centro Studi Dinamiche Complesse (CSDC), University of Florence, Sesto F.no Florence, 50019 Florence, Italy; 3Department of Environmental Microbiology and Biotechnology, Institute of Microbiology, Faculty of Biology, University of Warsaw, 02-096 Warsaw, Poland; mikolaj.dziurzynski@biol.uw.edu.pl; 4Magnetic Resonance Center (CERM) and Department of Chemistry “Ugo Schiff”, University of Florence, via Sacconi 6, Sesto Fiorentino, 50019 Fiorentino, Italy; turano@cerm.unifi.it (P.T.); veronica.ghini@unifi.it (V.G.); 5Consorzio Interuniversitario Risonanze Magnetiche MetalloProteine (CIRMMP), via Sacconi 6, Sesto Fiorentino, 50019 Fiorentino, Italy; 6Dipartimento di Scienze Chimiche, Complesso Universitario Monte Sant’Angelo, 80126 Napoli, Italy; marzia.calvanese@gmail.com (M.C.); andrea.colarusso@unina.it (A.C.); concetta.lauro@unina.it (C.L.); erparril@unina.it (E.P.); tutino@unina.it (M.L.T.); 7Istituto Nazionale Biostrutture e Biosistemi—I.N.B.B., Viale Medaglie d’Oro, 305-00136 Roma, Italy

**Keywords:** CDKL5, genome-scale metabolic modelling, protein production

## Abstract

hCDKL5 refers to the human cyclin-dependent kinase like 5 that is primarily expressed in the brain. Mutations in its coding sequence are often causative of hCDKL5 deficiency disorder, a devastating neurodevelopmental disorder currently lacking a cure. The large-scale recombinant production of hCDKL5 is desirable to boost the translation of preclinical therapeutic approaches into the clinic. However, this is hampered by the intrinsically disordered nature of almost two-thirds of the hCDKL5 sequence, making this region more susceptible to proteolytic attack, and the observed toxicity when the enzyme is accumulated in the cytoplasm of eukaryotic host cells. The bacterium *Pseudoalteromonas haloplanktis* TAC125 (PhTAC125) is the only prokaryotic host in which the full-length production of hCDKL5 has been demonstrated. To date, a system-level understanding of the metabolic burden imposed by hCDKL5 production is missing, although it would be crucial for upscaling of the production process. Here, we combined experimental data on protein production and nutrients assimilation with metabolic modelling to infer the global consequences of hCDKL5 production in PhTAC125 and to identify potential overproduction targets. Our analyses showed a remarkable accuracy of the model in simulating the recombinant strain phenotype and also identified priority targets for optimised protein production.

## 1. Introduction

The possibility to heterologously express and purify specific recombinant proteins in large amounts permits their biochemical characterisation, the development of commercial goods and their use in industrial processes. With the development of recombinant insulin and its production in *Escherichia coli* in the 1980s [1], a multi-billion dollar market was launched, leading to current large-scale applications that are nowadays capable of releasing products ranging from protein biologics to industrial enzymes [2]. Ideally, the practical steps that lead to recombinant protein production are pretty straightforward and include the identification of the gene of interest, its cloning into an expression vector, its transformation into the host of choice, the induction of protein synthesis and its final purification and characterisation [3]. The intrinsic complexity of biological systems, however, usually poses problems down the pipeline of bacterial heterologous protein production. Indeed, as a consequence of the induction of the production of the foreign protein, the biochemistry and physiology of the host may be dramatically altered. The numerous physiological changes that may occur often lower the amount of the target foreign protein that is produced and eventually recovered from the recombinant organism [4]. In bacteria, high levels of recombinant protein production frequently lead to an impact on host cell metabolism; this is usually detectable through growth retardation and is generally known as ‘metabolic burden’ [5]. This additional metabolic load on the microbial chassis has been defined as the portion of a host cell’s resources—either in the form of energy, such as ATP or GTP, or raw materials, such as amino acids—that is required to maintain and express foreign DNA, as either RNA or protein, in the cell [4]. In *E. coli*, for example, the overexpression of an unnecessary protein results in a linear decrease in the growth rate, with the zero-growth limit occurring when the overexpressed protein occupies a mass fraction equal to 1-$\phi_{fixed}$, with $\phi_{fixed}$ representing the growth-rate invariant fraction of the proteome [6]. There are many factors contributing to the emergence of this burden on growing cells that, at the same time, express a heterologous protein. These mainly include the transcription, translation and folding of the foreign protein [5,7,8] and the processes associated with plasmid maintenance, expression and amplification [7,8]. In addition, the expression of recombinant proteins may induce a system-level stress response that downregulates key metabolic pathway genes, leading to a decline in cellular health and feedback inhibition of both growth and protein expression [9]. Finally, from an energetic perspective, the expression of a foreign protein in a cell may use a significant fraction of its metabolic resources and precursors, removing them away from its central metabolism and placing a metabolic drain on the host [4]. Thus, upon protein production induction, an overall cellular reprogramming has to occur in order to ensure an adequate supply of energy and charged amino acids to the process of protein synthesis [9]. The identification of these system-level adjustments following heterologous protein production requires the use of computational representations of microbial metabolism that are able to consider the entire cellular metabolic network. In addition, these computational models may help identify the most suitable approaches to getting to target (protein) overproduction. Indeed, it has been recently acknowledged that the most innovative approach currently available to improving the yield of recombinant proteins, while minimising wet-lab costs, relies on the combination of in silico studies to reduce the experimental search space [10]. Among all the available in silico approaches, genome-scale metabolic models (GEMs) offer the possibility to predict a cellular phenotype from a genotype under certain environmental conditions and, importantly, to identify possible metabolic targets to improve the production of valuable compounds, while ensuring sufficiently high growth rates [11,12,13]. GEMs can also be used for descriptive purposes, including the identification of specific metabolic rewiring strategies following external perturbations and/or a nutrient switch [14,15]. Thus, not surprisingly, GEMs have been extensively exploited in the context of recombinant protein production, mostly with the aim of optimising either the cultivation conditions or the strain genetic background for improved recombinant protein production [16,17,18].

Although *E. coli* is arguably the bacterium of choice for the production of recombinant proteins, the emergence of a novel bacterial chassis is an important fact, especially considering the possible unique properties of their physiology and metabolism and the practical applications in which they are expected to outperform other microbial platforms [19]. Among them, *Pseudoalteromonas haloplanktis* TAC125 (PhTAC125), the first Antarctic bacterium in which an efficient gene expression technology was established [20], is particularly promising for a number of reasons. Firstly, several generations of cold-adapted gene expression vectors allow one to produce recombinant proteins either by constitutive or by inducible systems and to address the product towards any cell compartment or to the extracellular medium. Secondly, the development of synthetic media and efficient fermentation schemes allows upscaling the recombinant protein production in automatic bioreactors. Finally, the recently reported possibility to produce proteins within a range of temperature from 15 to −2.5 °C enhances the chances to improve the conformational quality and solubility of recombinant proteins. Up to now, PhTAC125 has been used for the production of several recombinant proteins, such as a psychrophilic β-galactosidase, *S. cerevisiae* α-glucosidase, human nerve growth factor and the lysosomal enzyme α-galactosidase A (hGLA) [21,22,23].

Recently, PhTAC125 was found to be a potential chassis for the production of human CDKL5 (hCDKL5). hCDKL5 is a cyclin-dependent-like protein kinase abundantly expressed in the brain, and it exerts its function in different neuron districts, such as the nucleus, the cytoplasm and the synaptosome. Mutations in the X-linked *cdkl5* gene often end up in the enzyme absence or in the production of loss-of-function variants, and both conditions are causative of hCDKL5 deficiency disorder (CDD), a rare and severe neurodevelopmental disorder for which no cure is available [24]. Recently, a protein replacement therapy was suggested, consisting of the administration of protein transduction domain (TAT)-fused hCDKL5 (TAT-CDKL5). When injected in *cdkl5*-knockout mice, TAT-CDKL5 was able to rescue many anatomical and behavioural deficits [25]. The translation of this promising therapeutic approach to clinics needs the large-scale recombinant production of TAT-CDKL5. However, full-length human CDKL5 is a difficult-to-produce enzyme for two main reasons:(i)Almost two-thirds of its sequence is predicted to be intrinsically disordered, and the lack of a precise 3D structure makes this region more susceptible to proteolytic attack by host-encoded proteases.(ii)The cytoplasmic accumulation of the enzyme in eukaryotic cells is associated to considerable toxicity, and the only permissive production strategy is its extracellular secretion, often accompanied with unwanted glycosylation [26]. PhTAC125 is the only prokaryotic cell factory in which full-length hCDKL5 production has been demonstrated, and the implementation of its efficient production process is the obligatory step towards any possible application (Calvanese et al., 2021, in press).

In this work, we modelled the heterologous production of the hCDKL5 protein in the bacterium PhTAC125. The genome-scale model of the recombinant strain was based on its original formulation [27] and further refined/updated and constrained with experimental data on hCDKL5 production and substrate consumption. This recombinant model was then used to study the global metabolic consequences of the induction of hCDKL5 production as well to identify potential targets for its overproduction.

## 2. Results and Discussion

### 2.1. An Updated Metabolic Reconstruction of PhTAC125

The latest version of the iMF721 metabolic model of *P. haloplanktis* TAC125 [27] was updated to be compatible with the current Systems Biology Markup Language Level 3 Version 2 Core specification [28] extended with the Flux Balance Constraints version 2 package specification [29]. The update was conducted using the *libsbml* Python library. It covered appropriate objective function declaration, compartment redefinition, model definition annotation with SBO terms, extension of species definitions with chemical formulas, update of gene names with the newest version of the *P. haloplanktis* genome and various minor syntax changes. The update increased the iMF721 Memote Total Score from 30% to 78% (Memote reports are available at https://github.com/mdziurzynski/tac125-metabolic-model, accessed on 16 July 2021). Additionally, we used BOFdat [30] to revise the original definition of the biomass composition in iMF721 using available experimental data. We also used the revised genome sequence of *P. haloplanktis* [31] and a compendium of transcriptomics data from previously published works [32] to improve the formulation of the biomass assembly reaction originally proposed [27]. After updating the model, we checked whether it could quantitatively reproduce growth phenotypes, as done by the original metabolic reconstruction. Growth simulations on defined media revealed an overall accuracy that matched the one of the original iMF721 reconstruction (Appendix A). This updated version of the model was referred to as iMF721_v2 in subsequent sections and is available at https://github.com/mfondi/CDKL5_recombinant_production, accessed on 16 July 2021.

### 2.2. CDKL5 Production in Controlled Growth Conditions

Human CDKL5 was recombinantly expressed as an N-terminally His-tagged engineered construct to allow for easy Western blot detection and quantification. Its gene was expressed under the control of an IPTG-regulatable promoter [33] cloned in a high-copy-number plasmid, named pB40_79C-CDKL5 (average copy number equal to 100, manuscript in preparation), in a mutant version of PhTAC125-KrPl LacY+ capable of fast IPTG internalisation [33]. hCDKL5 synthesis was induced in the late exponential phase with 5 mM IPTG at 15 °C in bacteria grown in GG medium [34] for 8 h. Total production of the target protein was estimated to be 5.2 mg/L of the culture by Western blot using a commercial His-tagged calibrator with a similar MW as hCDKL5.

### 2.3. Estimation of Average hCDKL5 Production Flux and Nutrients Uptake Rates

Here, we computed the actual (average) production and growth rates from the experimental data. As for hCDKL5 (molecular weight 128,082.77 mg mmol^−1^), after 8 h, a total amount of 5.2 mg (for 1 L of culture) was obtained. After the same amount of time, the OD of the culture was measured to be 2.55, which, when multiplied by 0.74 (i.e., the factor for converting PhTAC125 OD to grams of biomass [34]), corresponds to 1.887 g of cell dry weight (CDW). Putting everything together, we can compute the average production flux of hCDKL5 as follows:[(5.2 mg/8 h)/128,082.77 mg mmol^−1^]/1.887 g_CDW_ = 2.7 × 10^−6^ mmol/g_CDW_ h^−1^

The average growth rate for the recombinant strain across the 8 h period was computed using initial and final OD values:(ln2.55 − ln0.94) ÷ 8 = 0.125 h^−1^

The same calculation led to an average growth rate of 0.169 h^−1^ for the WT strain. According to these data, the production of hCDKL5 imposes an overall burden on growing PhTAC125 cells, which leads to a 26% reduction in biomass production in the hCDKL5 strain (Figure 1A).

At this point, the only parameters that are missing to fully characterise the CDKL5 production dynamics are the uptake rates for glutamate and gluconate when they represent the only C sources on a minimal medium. To calculate these, we set up an ad hoc experiment (see Section 3 (Materials and Methods) and Appendix A) that revealed an uptake rate of 0.35 and 0.66 mmol/g_CDW_ h^−1^ for glutamate and gluconate, respectively.

### 2.4. Recombinant Model Construction to Account for hCDKL5 Production

We then extended iMF721_v2 to include heterologous hCDKL5 production (leading to iMF721_v2_CDKL5 reconstruction (see Appendix A). The processes taken into account are (i) synthesis of the pB40 plasmid and (ii) synthesis of hCDKL5 mRNA and its translation into the corresponding protein sequence. As hCDKL5 is not secreted by PhTAC125, no energy-dependent hCDKL5 secretion reaction was added to the model. A plasmid copy number (Pcn) of 100 was used for pB40 because the latter is a high-copy-number plasmid. The reaction included in the metabolic network of PhTAC125 representing the synthesis of pB40 is the following:21H_2_0 + 21ATP + 57dATP + 43dGTP + 43dCTP + 57dTTP -> pB40 + 21ADP + 21Pi + 21H

The stoichiometric coefficients for dATP, dGTP, dCTP and dTTP were determined according to the GC composition of the pB40 plasmid. The ATP requirement for the synthesis of the pB40 plasmid was estimated based on the amount of ATP required for the synthesis of the chromosomal DNA, as previously described [16,18]. The obtained value (0.21) was multiplied by 100, the estimated copy number of pB40. Finally, pB40 was included in the biomass reaction of the model to account for the burden of the plasmid on the overall physiology of the cell. The stoichiometric coefficient of pB40 was again derived from the stoichiometric coefficient of chromosomal DNA in the biomass assembly reaction of iMF721_v2. This was done using the following proportion: 3850,272:0.001608 = 8166:100X, where the first, second, third and fourth terms represent the size (in bp) of the PhTAC125 genome, the stoichiometric coefficient for DNA in the original formulation of the PhTAC125 biomass reaction, the length of the pB40 plasmid and the (unknown) actual stoichiometric coefficient for the 100 copies of the plasmid, respectively. This calculation led to a stoichiometric coefficient for pB40 of 0.000341. Concerning the reaction for hCDKL5 synthesis, this was formalised as follows:59Ala[c] + 6 Cys[c] + 63 Asp[c] + 76 Glu[c] + 32 Phe[c] + 72 Gly[c] + 49 His[c] + 39 Ile[c] + 84 Lys[c] + 101 Leu[c] + 20 Met[c] + 59 Asn[c] + 80 Pro[c] + 53 Gln[c] + 76 Arg[c] + 140 Ser[c] + 56 Thr[c] + 41 Val[c] + 6 Trp[c] + 32 Tyr[c] + 2288 atp[c] + 2286 gtp[c] -> cdkl5[c] + 2288 amp[c] + 2286 gdp[c] + 4574 Pi[c]’,
where the stoichiometric coefficients for the amino acids were based on the composition of the protein sequence and the amount of ATP was computed considering the requirement of four ATP molecules for each amino acid added to the protein [35]. As said above, since hCDKL5 is not exported from the cell in vivo, no active transport reaction was included in the model.

At this point, we constrained this iMF721_v2_CDKL5 reconstruction with experimental data to build two further models, i.e., a *wt* model and a recombinant model (named *recomb* for brevity). More specifically, we constructed:A *wt* model by constraining the iMF721_v2_CDKL5 reconstruction with glutamate/gluconate uptake rates to the values experimentally determined and setting the biomass assembly reaction as the BOF of the modelA *recomb* model by constraining the iMF721_v2_CDKL5 reconstruction with glutamate/gluconate uptake and growth rates to the values experimentally determined and setting the hCDKL5 production reaction as the BOF of the model

These two models were used for all the simulations described below. The schematic representation of the computational steps leading to the two models is reported in (Appendix A).

### 2.5. The PhTAC125 Recomb Model Accurately Simulates hCDKL5 Production

To account for the predictive capability of PhTAC125 reconstruction in the context of hCDKL5 production, we computed growth and hCDKL5 production rates in the *wt* and *recomb* models.

As said above, the *wt* model was obtained by setting the lower bound of glutamate and gluconate uptake reactions to 0.35 and 0.66 mmol/g_CDW_ h^−1^, respectively, and performing an FBA simulation using biomass production as the objective function. This *wt* model predicted a growth rate of 0.119 h^−1^, which closely resembles the one experimentally measured (Figure 1B, “WT”). Afterwards, to generate the *recomb* model, we maintained the same boundaries for the glutamate and gluconate reactions and constrained the growth rate to 74% of the optimal one predicted by the model (74% of 0.119 h^−1^) and optimised for hCDKL5 production (Figure 1A). The simulations using this *recomb* model returned a hCDKL5 production flux of 2.67 × 10^−6^ mmol/g DCW h^−1^, which accurately resembles the one measured experimentally (2.7 × 10^−6^ mmol/g DCW h^−1^) (Figure 1C). A production envelope analysis correctly revealed that hCDKL5 production and biomass production compete for a common pool of nutrients and allowed us to sketch the current trade-off between these two cellular objectives (Figure 1D).

These data indicate that when constrained with experimental data, the *recomb* model is capable of providing a stoichiometrically reliable representation of hCDKL5 production in PhTAC125.

### 2.6. PhTAC125 Metabolic Rewiring Following hCDKL5 Induction

To explore the extent of PhTAC125 metabolic network rewiring upon the induction of hCDKL5 synthesis, we then analysed the differences in flux distributions between *wt* and *recomb* models. As expected, running an FBA simulation on the two models, we found a different number of flux-carrying reactions, with the recomb model showing a higher number of *core* reactions (491 vs. 484). However, since an FBA solution may not be unique (i.e., alternative flux distributions may still lead to an equally optimal solution), we used flux variability analysis (FVA) to assess the set of *core* reactions in each of the simulations (see Materials and Methods). A set of 84 *core* reactions was shared by the *wt* and *recomb* models. This set of reactions represented 74% and 97%, respectively, of the *core* reactions of the two models (i.e., the set of reactions remaining after removing the set of reactions showing a large variability range). Within this set, we identified 12 reactions (11 of them were gene encoded) shared by both models but that showed an increased flux in the *recomb* vs. the *wt* model (Table 1). The 11 gene-encoded reactions included the reactions involved in histidine biosynthesis and an ammonia transporter. The histidine biosynthetic reactions covered the entire pathway, i.e., from 5-phosphoribosyl 1-pyrophosphate (PRPP) to histidine. The higher flux predicted in the histidine biosynthetic pathway of the *recomb* model vs. the *wt* model can be explained by the different amino acid composition of recombinant protein with respect to the native PhTAC125 proteome (Figure 2). Indeed, as the abundance of this amino acid is double in hCDKL5 with respect to the PhTAC125 proteome, precursors used to produce histidine in the *recomb* model will be drained faster than in the *wt* model and fluxes around those precursors are expected to be significantly altered [36].

### 2.7. Finding the Optimal Growth Medium

We then sought to identify potential carbon sources whose inclusion in the original, optimised medium could boost the production of hCDKL5. To this purpose, we selected all the transport reactions present in the iMF721_v2 metabolic reconstruction and created a list including the transported compounds. We considered PhTAC125 as capable of taking up these compounds inside the cell because its genome encodes the corresponding transporters. We then performed one simulation for each of these compounds, adding it to the defined medium used during the previous simulations (Schatz salts plus glutamate and gluconate; see Materials and Methods), constraining the growth rate to the experimentally determined value and using hCDKL5 production as the objective function. In these simulations, the uptake rate of the extra carbon source was arbitrarily set to 0.5 mmol/g_CDW_ h^−1^. We then estimated the effect of the amended carbon source by computing the ratio between the hCDKL5 production flux in the new carbon source and the original one (i.e., with no amendments) and selected the first 30 compounds in the list (Figure 3).

The three most promising compounds identified in our analysis were amylotriose, maltose and mannitol. The first catabolic steps of these three compounds led to the formation of d-glucose (amylotriose and maltose) or d-fructose in PhTAC125, thus suggesting that the strengthening of sugar metabolism might the primary effect of adding these compounds to the growth medium of the recombinant strain and one of the possible ways to increase hCDKL5 production. To better address this point, we further investigated which part of the PhTAC125 metabolic network is specifically affected by the amendment of the best-performing nutrients to the growth medium. We thus checked which reactions increased their flux in the recombinant model growing in GG medium plus amylotriose compared to the same model grown in simple GG medium (Table 2). This list of reactions was filtered by removing those (*non-core*) reactions showing more than 30% variation between their minimum and maximum fluxes during an FVA, as described in Materials and Methods. Overall, we found 15 gene-encoded reactions displaying an increased flux in this condition: the majority of them (11) were involved in histidine biosynthesis; 3 in phenylalanine, tyrosine and tryptophan biosynthesis; and 1 in riboflavin metabolism. We found a similar scenario (i.e., the same involved pathways) for the other top four nutrients (maltose, mannitol, thymidine and galactose), with a majority of histidine and phenylalanine metabolism-related enzymes displaying an increased flux in the amended medium. Additional pathways that might be affected by these nutrients include nicotinate and nicotinamide metabolism (found when simulating the amendment of thymidine) and galactose metabolism (found when simulating the amendment of galactose).

Taken together, these results suggest that the main effect of adding extra nutrients to the medium would be an increased availability of histidine molecules inside the cell, which, in turn, would result in an improved production rate of hCDKL5. Again, this can be explained by the different histidine content of the overall PhTAC125 proteome and of the hCDKL5 sequence (Figure 2). The enzymes involved in phenylalanine, tyrosine and tryptophan biosynthesis that appear to increase their flux in the tested conditions include those responsible for the generation of 5-phospho-alpha-D-ribose 1-diphosphate (PRPP), which is a key pentose phosphate pathway (PPP) intermediate for purine, pyrimidine and histidine biosynthesis. Its connection to increased hCDKL5 production might thus be double: on the one side, it could fuel histidine biosynthesis for the reasons described above; on the other side, it could facilitate the synthesis of purines and pyrimidines required by plasmid replication and transcription during heterologous protein expression.

### 2.8. Finding Hypothetical Targets for hCDKL5 Overproduction

We then used the model to predict possible targets to improve the production of hCDKL5 in *P. haloplanktis* TAC125. We focused our attention on the use of the well-established FSEOF algorithm [37]. Briefly, FSEOF scans all the fluxes in the reconstruction and identifies the increasing ones when the flux towards product formation is set (enforced) as a further constraint during FBA. The reactions identified by FSEOF are primary overexpression targets that may lead to improved synthesis of the desired target (hCDKL5 in our case). By applying FSEOF, as described in Materials and Methods, we identified 70 target gene-encoded reactions whose overexpression may lead to improved target production. The complete list of these reactions is available in Appendix A. The top 10 target reactions identified by FSEOF are shown in Table 3. The first reaction in the list is represented by rxn05937, catalysing the formation of NADPH from NADP and reduced ferredoxin. Forcing the flux through this reaction would allow increasing the overall NADPH pool of the cell, and this has been widely recognised as an important factor in the process of heterologous protein production in microorganisms [38]. Reduced ferredoxin necessary for the production of NAPH might be provided by l-glutamateferredoxin oxidoreductase (Table 3), catalysing the conversion of L-glutamate to L-glutamine with the reduction of ferredoxin. Reactions belonging to the Entner–Doudoroff (ED) branch of the PPP are also high-ranking overexpression targets according to FSEOF (Table 3, Figure 4A). These include the three enzymes catalysing the conversion to d-glucono-1,5-lactone 6-phosphate to pyruvate and d-glyceraldehyde 3-phosphate (encoded by *agaI*, *edd* and *eda*). Overall, the degradation of one molecule of glucose through this pathway, as opposed to classical PPP leading to ribose-5P, leads to lower amounts of reducing equivalents (one NADPH produced instead of two) but ensures a greater and balanced production of precursors (namely pyruvate and glyceraldehyde-3P (G3P)) that can be used both to fuel the TCA cycle and for amino acid biosynthesis [39]. Indeed, it is known that the ED pathway, as a variant glycolysis pathway, produces equal amounts of G3P and pyruvate, and this superior stoichiometric feature makes the ED pathway a preferable route for precursor supply [40]. Importantly, targets within these metabolic pathways (i.e., ED and PPP in general) have been identified in other works aimed at identified optimisation production strategies [18,19,39,41]. Most of the other reactions identified by the FSEOF algorithm are involved in the metabolism of amino acids. In particular, our simulations suggest that the production of hCDKL5 might be improved by redirecting the catabolism of glutamate towards the production of aspartate (through the action of l-aspartate-2-oxoglutarate aminotransferase) and its subsequent conversion to 4-phospho-L-aspartate and l-aspartate-4-semialdehyde (Figure 4B), catalysed by ATPL-aspartate-4-phosphotransferase and l-aspartate-4-semialdehyde: NADP + oxidoreductase, respectively. l-aspartate-4-semialdehyde, in particular, serves as a substrate for the biosynthesis of many amino acids, including lysine, threonine and glycine (Figure 4B). Finally, our FSEOF simulation identified the enzyme serine O-acetyltransferase (catalysing the formation of serine from CoA and O-acetyl-l-serine) as a likely hCDKL5 overproduction target. Looking at the unbalanced distribution of S residues in the sequence of hCDKL5 with respect to the one of the PhTAC125 genome (Figure 2), it can be hypothesised that the meaning of this latter finding resides in the necessity to increase the production of serine to cope with the higher request of this amino acid following the induction of CDKL5 production.

Further, to provide a general view of the reactions identified as potential overexpression targets by FSEOF, we grouped them according to their corresponding metabolic pathway (Figure 4C). In line with the results illustrated above, 6 pathways (out of 10 with more than two reactions included) were representatives of amino acid metabolism, with 4 of them appearing in the top five pathways (i.e., Val/Leu/Ile, Phe/Tyr/Trp and Lys biosynthesis and His metabolism). In addition to amino acid metabolism, the other pathways represented were urea and amino group metabolism, glycolysis, the PPP and purine metabolism.

## 3. Materials and Methods

### 3.1. Bacterial Strains and Conjugation Experiments

The pB40_79C-CDKL5 plasmid was mobilised from *E. coli* S17-1(λpir) to KrPL LacY+ [33] through standard conjugation techniques [41]. *E. coli* S17-1(λpir)—a strain possessing *mob* and *tra* genes for plasmid mobilisation [42]—was routinely grown in LB (10 g/L of bacto-tryptone, 5 g/L of yeast extract, 10 g/L of NaCl) at 37 °C with the supplementation of 34 μg/mL of chloramphenicol, if needed, for plasmid selection. KrPL LacY+, a *P. haloplanktis* TAC125 strain engineered for improved IPTG uptake [33], was grown at 15 °C in TYP (16 g/L of bacto-tryptone, 16 g/L of yeast extract, 10 g/L of NaCl) for conjugational experiments and initial pre-inocula. Recombinant KrPL LacY+ was selected with 25 and 12.5 μg/mL of chloramphenicol in liquid and solid media, respectively. Solid LB and TYP broths were prepared by the addition of 15 g/L of agar.

### 3.2. hCDKL5 Production

The pB40_79C-CDKL5 plasmid allows the IPTG-inducible expression of a PhTAC125 codon optimised gene coding for an engineered variant of human CDKL5 isoform 1. The translated protein possesses tandem His-Sumo [43] and Tatk [25,44] N-terminal tags and a C-terminal 3xflag. The whole 1144 aa sequence was expressed as a cytosolic protein from the pB40 plasmid, which is characterised by an average copy number of 100 (manuscript in preparation). For recombinant gene expression, KrPL LacY+ was cultivated at 15 °C in a 100 mL Erlenmeyer flask containing 20 mL of GG medium [34]: 10 g/L of L-glutamic acid monosodium salt monohydrate, 10 g/L of gluconic acid sodium salt, 10 g/L of NaCl, 1 g/L of NH_4_NO_3_, 1 g/L of KH_2_PO_4_, 0.2 g/L of MgSO_4_·7H_2_O, 5 mg/L of FeSO_4_·7H_2_O and 5 mg/L of CaCl_2_ 2H_2_O (pH 7.8). After inoculating at 0.10 OD600, bacterial growth was followed for 13 h and the recombinant gene expression triggered at 1.00 OD600 with 5 mM IPTG. Eight hours after induction, the bacterial cells were harvested by centrifugation (4 °C, 4000× *g*, 20 min) when they reached 2.55 OD600. To check and estimate hCDKL5 intracellular production at the end of the culture, bacterial pellets equivalent to 1.00 OD600 were resuspended in 60 μL of Laemmli Buffer 4× and denatured at 90 °C for 20 min. Denatured cellular extracts equivalent to 1/120 OD600 were loaded onto a 7.5% precast Mini-Protean TGX (BioRad Laboratories, Hercules, CA, USA) and resolved by SDS-PAGE. Known amounts of His-Neuropilin (110 kDa; Immunological Sciences, Rome, Italy) were loaded onto adjacent lanes to develop a calibration curve. Then, separated proteins were transferred to a PVDF membrane using a semi-dry system, and His-tagged proteins (hCDKL5 and His-Neuropilin) were detected with an HRP-conjugated anti-His antibody (1:2000; Sigma-Aldrich) using the enhanced chemiluminescence (ECL) kit (BioRad, Hercules, CA, USA) and a ChemiDoc MP Imaging System (BioRad, Hercules, CA, USA). Quantitative analyses of blotted hCDKL5 and His-Neuropinilin were carried out using Image Lab software (BioRad, Hercules, CA, USA), and the volumetric yield was derived considering the final biomass concentration (OD600: 2.55).

### 3.3. Glutamate and Gluconate Consumption Experiment

*Ph*TAC125 bacterial culture was grown in GG medium modified so to contain 5 g/L of l-glutamic acid monosodium salt monohydrate and 5 g/L of d-gluconic acid sodium salt in a stirred tank reactor with a 3 l fermenter (Applikon, Schiedam, The Netherlands) with a working volume of 1.5 L. The bioreactor was equipped with standard pH, pO_2_, level and temperature sensors for bioprocess monitoring. Culture was carried out at 15 °C for 30 h under aerobic conditions (45% dissolved oxygen). Next, 1 mL samples for metabolomic analysis were collected during growth and centrifuged at 1300 rpm for 20 min at 4 °C. After centrifugation, supernatants were recovered, filtered through membranes with a pore diameter of 0.22 µm and stored at −80 °C.

### 3.4. Metabolomic Data

Metabolomic data on cell growth media were obtained by 1H nuclear magnetic resonance (NMR) spectroscopy. The supernatant samples were thawed at room temperature. Next, 540 μL of each sample was added with 60 μL of potassium phosphate buffer (1.5 M K_2_HPO_4_, 100% (*v*/*v*) 2H_2_O, 10 mM sodium trimethylsilyl [2,2,3,3−2H_4_] propionate (TMSP), pH 7.4). The mixture was transferred into 5 mm NMR tubes for subsequent analysis.

Spectral acquisition and processing were performed according to standard procedures [45,46]. One-dimensional (1D) 1H NMR spectra were recorded using a Bruker 600 MHz spectrometer (Bruker BioSpin Gmbh, Rheinstetten, Germany) operating at 600.13 MHz proton Larmor frequency and equipped with a 5 mm PATXI 1H-13C-15N and 2H-decoupling probe, including a z-axis gradient coil, automatic tuning and matching and an automatic and refrigerate sample changer (SampleJet, Bruker BioSpin Gmbh, Rheinstetten, Germany). A BTO 2000 thermocouple served for temperature stabilisation at the level of ~0.1 K at the samples. Before measurement, samples were kept for 5 min inside the NMR probe head for temperature equilibration at 300 K.

NMR spectra were acquired with water peak suppression using the 1D standard NOESY pulse sequence (128 scans, 65,536 data points, spectral width of 12,019 Hz, acquisition time of 2.7 s, relaxation delay of 4 s and mixing time of 0.01 s).

The raw data were multiplied by 0.3 Hz exponential line broadening before applying Fourier transformation. Transformed spectra were automatically corrected for phase and baseline distortions. All spectra were then calibrated to the reference signal of TMSP at δ = 0.00 ppm using TopSpin 3.5 (Bruker BioSpin Gmbh, Rheinstetten, Germany).

The signals deriving from glutamate and gluconate were assigned using an internal NMR spectral library of pure organic compounds; matching between the present NMR spectra and the NMR spectral library was performed using AssureNMR software (Bruker BioSpin Gmbh, Rheinstetten, Germany). Their concentrations were calculated by integrating the corresponding signals in the defined spectral range using a home-made R 3.0.2 script.

### 3.5. PhTAC125 Genome-Scale Metabolic Reconstruction and Constraint-Based Simulations

The original *P. haloplanktis* TAC125 genome-scale metabolic reconstruction [27] was used as the starting point of the modelling procedures. This metabolic reconstruction was then updated and quality-checked, as described above, using BOFdat [30] (for the biomass reaction) and Memote [47] (model consistency evaluation).

The recently published genome sequence of *P. haloplanktis* TAC125 [31] was fed into BOFdat *DNA.py* script in order to generate the updated stoichiometric coefficients for As, Ts, Cs, and Gs. Similarly, a compendium of expression (RNAseq) data from previously published [32] datasets was fed into the BOFdat *RNA.py* code in order to generate revised and experimentally based stoichiometric coefficients for RNA building blocks.

Constraint-based simulation (e.g., FBA) were performed using COBRA Toolbox v3.0 [48] in MATLAB 2020b and using Gurobi as a solver. Overexpression targets were identified using the latest FSEOF version implemented in Raven [49] and selecting 100 iterations and a ratio coefficient of the optimal target reaction flux of 0.9. The codes used to run all the simulations are available at https://github.com/mfondi/CDKL5_recombinant_production, accessed on 26 July 2021.

### 3.6. Identification of Core Reactions

Flux variability analysis (FVA) was used to assess the relevance of each reaction when simulating growth and hCDKL5 production. The *fluxVariability* function of the COBRA toolbox was used for this purpose. The following procedure was applied (separately) to both *wt* and *recomb* models. First, an FBA optimisation was run on the model to predict the flux across each reaction. Afterwards, an FVA simulation with exactly the same constraints as the previous FBA simulation was performed and the flux range for each reaction stored. Then, for each of the two models, only those reactions satisfying the following criterion were labelled as *core* reactions:

with $sol_{wt/recomb}>0$:$$f_{min,\;FVA}>0.7\;\times \;sol_{wt/recomb}\;AND\;f_{max,\;FVA}\;<1.3\;\times \;sol_{wt/recomb}$$
with $sol_{wt/recomb}<0$:$$f_{max,\;FVA}<0.7\;\times \;sol_{wt/recomb}\;AND\;f_{max,\;FVA}>1.3\;\times \;sol_{wt/recomb}$$
with $sol_{wt}$, $f_{min,\;FVA}$ and $f_{max,\;FVA}$ representing the FBA solution, the lower FVA solution value and the upper FVA solution value, respectively. According to this strategy, in each simulation, only those (*core*) reactions displaying a flux value different from zero and with a narrow range of admissible flux (30%) during an FVA simulation were maintained, whereas those not satisfying this condition were considered unreliable and filtered away.

## 4. Conclusions

In this work, we combined experimental and computational approaches to characterising the production of recombinant hCDKL5 in the Antarctic marine bacterium *P. haloplanktis* TAC125. By constraining an updated genome-scale metabolic model of this bacterium with experimentally determined nutrient absorption rates, we were able to predict hCDKL5 production rates that matched those determined experimentally and to correctly estimate the burden (in terms of a reduction in biomass yield, about 25% compared to the *wt* strain) of protein production in this bacterium. Next, we used the model to describe the metabolic rewiring occurring in this bacterium upon the induction of hCDKL5 production and to identify possible overproduction strategies (both in terms of amendments to the original growth medium and in terms of overexpression targets). Despite the fact that each of these analyses highlighted specific pathways and/or targets that appear to be strongly connected to hCDKL5 production, common trends could be identified (e.g., the role played by reactions belonging to histidine metabolism and to the PPP). Taken together, our findings suggest that a possible future strategy for increasing the production of hCDKL5 in PhTAC125 may involve the overexpression of the target genes identified by the FSEOF algorithm and/or growth of the recombinant cells in media amended with one (or more) of the compounds that our simulations identified as the most promising in increasing the yield of the heterologous protein.

Work is currently in progress to experimentally verify both the hCDKL5 overproduction targets and the hypothetical amendments to the PhTAC125 growth medium capable of increasing the protein yield in silico.

## Figures and Tables

**Figure 1 metabolites-11-00491-f001:**
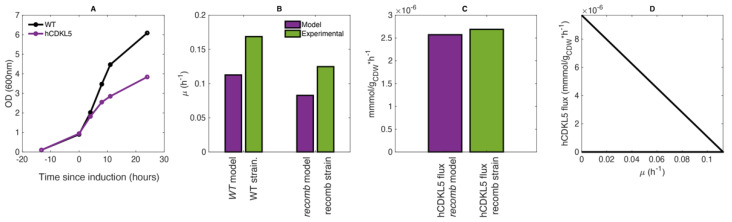
(**A**) Growth curves of WT and hCDKL5 strains, as experimentally determined. (**B**) Comparison between the model-predicted and measured growth rates in the wild-type strain. (**C**) Comparison between the measured hCDKL5 production rate in the recombinant strain and the one predicted by the model. (**D**) Production enveloper for hCDKL5.

**Figure 2 metabolites-11-00491-f002:**
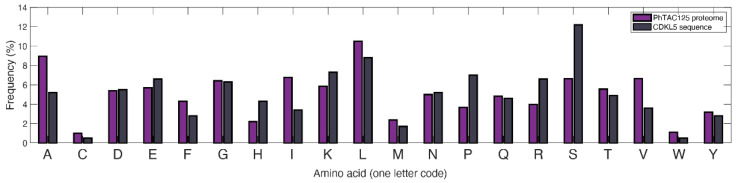
Difference in amino acid composition between the PhTAC125 proteome and CDKL5.

**Figure 3 metabolites-11-00491-f003:**
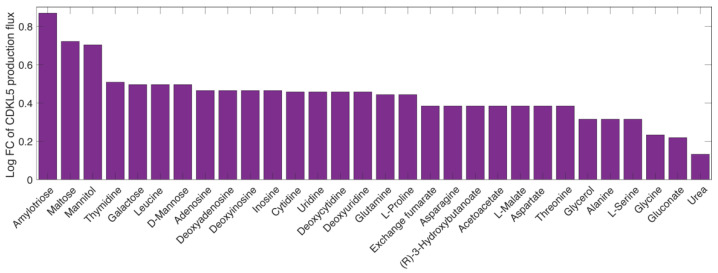
The effect of putative nutrients to be added to GG medium on the CDKL5 production flux. The *y*-axis indicates the log fold change of the CDKL5 production flux with respect to the *recomb* model grown on GG medium.

**Figure 4 metabolites-11-00491-f004:**
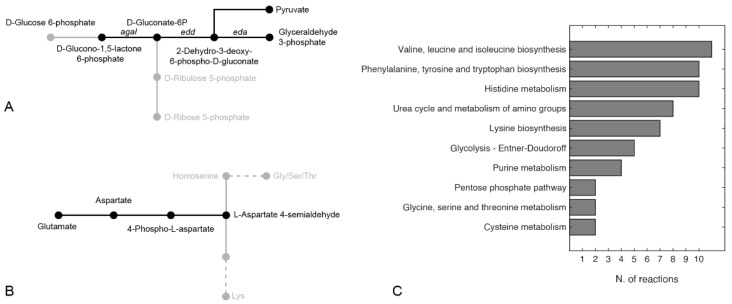
Pathways including reactions identified as potential overexpression targets by the FSEOF algorithm for (**A**) the Entner–Doudoroff pathway and (**B**) glutamate catabolism. Only pathways including 2 or more reactions are shown in (**C**).

**Table 1 metabolites-11-00491-t001:** Reactions showing an increased flux in the *recomb* vs. *wt* model simulations.

Reaction Model Code	Subsystem	Reaction Name
rxn00789	Histidine metabolism	1-(5-Phospho-d-ribosyl)-ATP:pyrophosphate phosphoribosyl-transferase
rxn00863	Histidine metabolism	l-histidinal:NAD + oxidoreductase
rxn02159	Histidine metabolism	l-histidinol:NAD + oxidoreductase
rxn02160	Histidine metabolism	l-histidinol-phosphate phosphohydrolase
rxn02320	Histidine metabolism	5-Amino-2-oxopentanoate:2-oxoglutarate aminotransferase
rxn02473	Histidine metabolism	d-erythro-1-(imidazol-4-yl)glycerol 3-phosphate hydrolyase
rxn02834	Histidine metabolism	Phosphoribosyl-ATP pyrophosphohydrolase
rxn02835	Histidine metabolism	1-(5-Phospho-d-ribosyl)-AMP 1,6-hydrolase
rxn03135	Histidine metabolism	Imidazole-glycerol-3-phosphate synthase
rxn03175	Histidine metabolism	N-(5′-phospho-d-ribosylformimino)-5-amino-1-
rxn05466	Ammonia transport	Ammonia transport via diffusion

**Table 2 metabolites-11-00491-t002:** Reactions showing an increased flux in the *recomb* model growing in GG medium amended with amylotriose.

Reaction Model Code	Subsystem	Reaction Name
rxn05466	Ammonium transporter	Ammonia transport via diffusion
rxn02160	Histidine metabolism	l-histidinol-phosphate phosphohydrolase
rxn00863	Histidine metabolism	l-histidinal:NAD + oxidoreductase
rxn02159	Histidine metabolism	l-histidinol:NAD + oxidoreductase
rxn02834	Histidine metabolism	Phosphoribosyl-ATP pyrophosphohydrolase
rxn03175	Histidine metabolism	N-(5′-phospho-d-ribosylformimino)-5-amino-1-
rxn02473	Histidine metabolism	d-erythro-1-(imidazol-4-yl)glycerol 3-phosphate hydro-lyase
rxn02508	Phenylalanine, tyrosine and tryptophan biosynthesis	N-(5-phospho-beta-d-ribosyl)anthranilate ketol-isomerase
rxn02320	Histidine metabolism	5-Amino-2-oxopentanoate:2-oxoglutarate aminotransferase
rxn02507	Phenylalanine, tyrosine and tryptophan biosynthesis	1-(2-Carboxyphenylamino)-1-deoxy-d-ribulose-5-phosphate
rxn00789	Histidine metabolism	1-(5-Phospho-d-ribosyl)-ATP:pyrophosphate phosphoribosyl-transferase
rxn03135	Histidine metabolism	Imidazole-glycerol-3-phosphate synthase
rxn00791	Phenylalanine, tyrosine and tryptophan biosynthesis	N-(5-phospho-d-ribosyl)anthranilate:pyrophosphate
rxn02835	Histidine metabolism	1-(5-Phospho-d-ribosyl)-AMP 1,6-hydrolase
rxn00392	Riboflavin metabolism	ATP:riboflavin 5′-phosphotransferase

**Table 3 metabolites-11-00491-t003:** Top 10 reaction targets predicted by the FSEOF algorithm.

Reaction Model Code	Subsystem	Reaction Name	Formula
rxn05937	NA	Ferredoxin:NADP+ oxidoreductase	NADP + H^+^ + reduced ferredoxin => NADPH + oxidised ferredoxin
rxn12822	Glyoxylate and dicarboxylate metabolism	l-glutamateferredoxin oxidoreductase (transaminating)	2 l-glutamate + 2 oxidised ferredoxin => 2-oxoglutarate + l-glutamine + 2 H^+^ + 2 reduced ferredoxin
rxn01477	PPP	6-Phospho-d-gluconate hydro-lyase (edd)	6-Phospho-d-gluconate => H_2_O + 2-keto-3-deoxy-6-phosphogluconate
rxn03884	PPP	2-Dehydro-3-deoxy-d-gluconate-6-phosphate d-glyceraldehyde-3-phosphate-lyase (eda)	2-Keto-3-deoxy-6-phosphogluconate => pyruvate + glyceraldehyde-3-phosphate
rxn01476	PPP	6-Phospho-d-glucono-1,5-lactone lactonohydrolase (AgaI)	H_2_O + 6-phospho-d-glucono-1-5-lactone => H^+^ + 6-phospho-d-gluconate
rxn00260	Alanine, aspartate and glutamate metabolism	l-aspartate2-oxoglutarate aminotransferase	2-Oxoglutarate + l-aspartate <= l-glutamate + oxaloacetate
rxn00337	Glycine, serine and threonine metabolism	ATPL-aspartate 4-phosphotransferase	ATP + l-aspartate => ADP + 4-phospho-l-aspartate
rxn01643	Glycine, serine and threonine-cysteine and methionine-lysine metabolism	l-aspartate-4-semialdehyde:NADP+ oxidoreductase (phosphorylating)	NADP + phosphate + l-aspartate-4-semialdehyde <= NADPH + 4-phospho-l-aspartate
rxn00285	Citrate cycle (TCA cycle)	Succinate-CoA ligase (ADP forming)	ATP + CoA + succinate => ADP + phosphate + succinyl-CoA
rxn00423	Cysteine and methionine metabolism	Serine O-acetyltransferase	Acetyl-CoA + l-serine <= CoA + O-acetyl-l-serine

## Data Availability

The data presented in this study are available as Appendix A and at https://github.com/mfondi/CDKL5_recombinant_production.

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
