# Peer review of "Modelling hCDKL5 Heterologous Expression in Bacteria"

_metabolites, 2021, doi:10.3390/metabo11080491_

Round 1
Reviewer 1 Report
No new physical insight is provided. The authors combine some well known reaction model and simulation already known to the field. I don't see any new physical insight in this study. Experimental design is poor. No direct evidence of optimized protein production as they have claimed. Some live cell protein expression imaging experiments are suggested.
Reviewer 2 Report
The manuscript presented modelling hCDKL5 heterologous expression combined with experimental data in P. haloplanktis TAC125. Before the paper could be accepted for publication in Metabolitesrs need make revisions as follow:
- Conclusion section should be revised, and discussion contents are suggested to delete.
- The strategy for increasing hCDKL5 heterologous expression should be presented clearly.
- Modification (phosphorylation) and folding of hCDKL5 heterologous expression must be discussed.
Round 2
Reviewer 1 Report
The authors answered all my queries and concerns. The revised version looks much better to me. I don't have any other concern.